# The effect of target transpulmonary driving pressure values on mortality in ARDS patients: A retrospective study based on the MIMIC-IV database

Na Liu[1◉], Qi Zhang[2,3◉], Huiyong Wang[3,4], Renshuang Ding[3,4], Xiaoyong Geng[5], Li Wang[6], Zhiyong Wang[2,3], Mingxing Fang[2,3,7]*

1 Department of Emergency, Forth hospital of Hebei Medical University, Hebei, China, 2 Department of Critical Care Medicine, Hebei Medical University Third Hospital, Hebei, China, 3 Critical Disease Data Analysis and Intelligent Diagnosis and Treatment Engineering Research Center of Hebei Province, Hebei Medical University Third Hospital, Hebei, China, 4 School of Information Science and Engineering, Hebei University of Science and Technology, Hebei, China, 5 Department of Cardiology, Hebei Medical University Third Hospital, Hebei, China, 6 Department of Oncology, Hebei Medical University Third Hospital, Hebei, China, 7 Department of Emergency and Internal Medicine, The People's Hospital of Tiemenguan, Tiemenguan, The Xinjiang Uygur Autonomous Region, China

◉ These authors contributed equally to this work.
* m18533112886@hebmu.edu.cn

## Abstract

### Background

This study examined the effect of target transpulmonary driving pressure on mortality in patients with Acute Respiratory Distress Syndrome, assessing how varying levels of transpulmonary driving pressure influence clinical outcomes.

### Methods

This retrospective study utilized data from the MIMIC-IV database to evaluate the relationship between transpulmonary driving pressure and mortality in Acute Respiratory Distress Syndrome. Associations between transpulmonary driving pressure levels and 28-day, ICU, and hospital mortality were analyzed. Propensity score matching was employed to balance covariates, while causal mediation analysis assessed whether peak airway pressure mediated the effect of transpulmonary driving pressure on mortality.

### Results

Among 4721 patients with Acute Respiratory Distress Syndrome, 295 received transpulmonary driving pressure targeting. The optimal transpulmonary driving pressure threshold was identified as 12.5 cmH$_2$O. Patients with transpulmonary driving pressure >12.5 cmH$_2$O had significantly higher 28-day, ICU, and hospital mortality,

**Data availability statement:** All relevant data are within the paper and its Supporting information files.

**Funding:** FMX was supported by fundings of the government of Hebei Province, China funded the provincial Medical Excellence Program and high-level talents funding Program of Hebei Province, China (grant number A20203005).

**Competing interests:** The authors have declared that no competing interests exist.

particularly in those with moderate to severe Acute Respiratory Distress Syndrome ($p < 0.05$). After propensity score matching, targeting transpulmonary driving pressure was associated with lower ICU mortality (HR 0.676, 95% CI 0.511–0.894, $p = 0.006$). Phenotypic analysis showed that elevated transpulmonary driving pressure was linked to worse outcomes in Phenotype-I(High Mechanical Power with Moderate Lung Compliance) and Phenotype-II (High Spontaneous Breathing with Better Lung Compliance), but not in Phenotype-III (Low Tidal Volume with Reduced Lung Compliance). Mediation analysis revealed that 7.0% of the mortality risk associated with transpulmonary driving pressure >12.5 cmH$_2$O was mediated through peak airway pressure.

## Conclusion

Transpulmonary driving pressure exceeding 12.5 cmH$_2$O is associated with higher mortality in Acute Respiratory Distress Syndrome patients, with peak airway pressure contributing to this effect.

## Introduction

Acute Respiratory Distress Syndrome (ARDS) is characterized by refractory hypoxemia and bilateral chest infiltration due to No-cardiogenic factors [1]. Mechanical ventilation serves as a primary life-support intervention for ARDS and plays a crucial role in reducing mortality [2]. Improper mechanical ventilation can worsen lung injury, leading to ventilator-induced lung injury (VILI) [3], which further increases mortality in ARDS patients. The lung-protective ventilation approach suggested by the ARDS Network includes low tidal volume ventilation (6 ml/kg), restricting plateau pressure (25–30 cmH$_2$O), and optimizing positive end-expiratory pressure (PEEP) [4]. However, current lung protective strategies, such as using lower tidal volume ventilation calculated by ideal body weight, fail to address the heterogeneity in disease manifestation among ARDS patients, which includes differences in disease severity and individual lung compliance [5,6].

Transpulmonary pressure (TPP), derived using esophageal pressure as a surrogate for pleural pressure, represents the true pressure exerted on alveoli and provides a more accurate measure of lung stress than airway pressure alone. A recent meta-analysis of randomized controlled trials conducted by the authors found that mechanical ventilation guided by transpulmonary pressure was associated with decreased mortality among patients with ARDS [7]. Pirrone's study found that an end-expiratory transpulmonary pressure above 0 cm H$_2$O facilitated the reopening of collapsed alveoli, reduced atelectasis, and improved oxygenation and lung compliance [8]. TPP-guided ventilation supports lung protection by preventing alveolar collapse, thereby contributing to reduced mortality in ARDS. However, excessive transpulmonary pressure may lead to alveolar overdistension, thereby increasing the risk of ventilator-induced lung injury, highlighting the need for careful pressure management [9–11].

Driving pressure (DP), defined as the difference between plateau pressure and PEEP during tidal breathing, is another key indicator of lung stress, reflecting both chest wall and alveolar contributions [12]. Transpulmonary driving pressure (TPDP), which isolates the alveolar component by accounting for chest wall compliance, may offer a more precise assessment of lung injury risk and ventilation optimization [13]. While studies have established a correlation between DP and mortality [13,14], the specific impact of targeted TPDP values on ARDS outcomes remains insufficiently investigated.

This study examines the relationship between TPDP values and mortality in ARDS patients using data from the MIMIC-IV database. By identifying optimal TPDP thresholds, we aim to determine how specific pressure targets might improve outcomes, particularly within different ARDS severity subgroups.

## Materials and methods

### Data resource

This retrospective cohort study was conducted using the Medical Information Mart for Intensive Care (MIMIC) database, an open-access clinical critical care dataset. MIMIC-IV was developed by the Computational Physiology Laboratory at the Massachusetts Institute of Technology in collaboration with Beth Israel Deaconess Medical Center (BIDMC) and Philips Medical. It includes comprehensive records of all patients admitted to the intensive care unit (ICU) at BIDMC between 2008 and 2019. One author (Zhang Q) successfully completed the "Data or Specimen Study" course and test under the (CITI PROGRAM and obtained permission to access the MIMIC database (NO. 9535772) [15]. Since all patient data in the database were anonymized, ethical approval was not required for this study.

### Patients selection

Patients meeting the following criteria were extracted from the MIMIC-IV database: (1) age ≥18 years, (2) diagnosed with ARDS according to the Berlin definition [16], and (3) received invasive mechanical ventilation for more than 24 hours.

The exclusion criteria were: (1) patients younger than 18 years and (2) those who underwent invasive mechanical ventilation for less than 24 hours. For patients with multiple ICU admissions, only data from the first admission were included.

### Variables extraction

Relevant variables were selected based on existing literature and related clinical studies.

The severity of illness: Sequential Organ Failure Assessment (SOFA score) [17–20] and Acute Physiology Score (APSIII).

Demographic characteristics: Age, height, weight, and Body Mass Index (BMI) [19,20].

Vital signs: Systolic Arterial Pressure (ABPs) [18], Mean Arterial Pressure (ABPm), Diastolic blood pressure (ABPd), and Central Venous Pressure (CVP).

Respiratory characteristics: Tidal Volume [17,19,20], Spontaneous Respiratory Rate, Total Respiratory Rate [19,20], Set Respiratory Rate, Positive end-expiratory pressure (PEEP) [17–20], Plateau Pressure, Peak pressure, Driving pressure, and Mechanical power.

Laboratory test: White Blood Cell (WBC) [20,21], Urea Nitrogen (BUN), Glucose, Creatinine [20,21], Lactate level (Lac) [18,20], Total Bilirubin, Direct Bilirubin, Albumin (ALB), Platelet Count, and Prothrombin Time (PT).

Blood gas analysis parameters: P/F ratio of the first day (day1P/F ratio), PH [10–13], Partial pressure of arterial oxygen ($PaO_2$) [17,19,20], Partial pressure of arterial carbon dioxide ($PaCO_2$) [17–20], $HCO_3^-$.

Outcome: ICU mortality, 28-day mortality, hospital mortality, mechanical ventilation hours, ICU stayday, 28d survival day, and hospital stayday.

Data with more than 35% missing values were excluded (S1 Fig). The average of all hospitalization data was used after removing outliers, defined as observations falling outside 1.5 times the interquartile range. Missing values were imputed using the Miss Forest model. All data were extracted using Structured Query Language (SQL) in Google Cloud BigQuery and PostgreSQL (v12.0).



### Transpulmonary driving pressure measurement

Driving pressure (ΔP) was calculated as the difference between plateau pressure and positive end-expiratory pressure, defined as ΔP = Pplat – PEEP. Transpulmonary driving pressure (TPDP) accounts for the difference between end-inspiratory and end-expiratory transpulmonary pressures.

### Statistical analysis

Data analysis was performed using R version 4.3.1, with statistical methods structured as follows:

### Descriptive statistics

Normality was assessed using the Shapiro-Wilk test (S1 Table). Continuous variables following a normal distribution are expressed as mean ± standard deviation (MD ± SD), with comparisons between two groups conducted using the t-test. No-normally distributed continuous variables are presented as median (interquartile range) [Median (IQR)], with comparisons performed using the Wilcoxon test. Categorical variables are expressed as percentages.

### Survival analysis

Survival curves were generated using the Kaplan-Meier method, and differences in survival status between groups were compared using the Log-rank test. The association between transpulmonary pressure and prognosis in ARDS patients was evaluated using univariate logistic regression and univariate Cox proportional hazards regression models. A $p$-value < 0.05 was considered statistically significant. Optimal cutoff values for continuous variables in survival analysis were determined using the surv_cutpoint() function from the survminer package, based on maximally selected rank statistics computed by the maxstat.test function.

### Propensity score matching

Propensity score matching (PSM) was applied to balance covariates, reduce confounding bias, and improve comparability between groups. Given that the data were derived from observational studies, potential confounding factors could introduce bias into the results. To address this, PSM was performed using the MatchIt package in R 4.3.1, employing the nearest neighbor matching method with a 1:1 ratio and a caliper of 0.05. Covariates included age, BMI, APSIII, SOFA score, diastolic and systolic arterial blood pressure (ABPd, ABPs), heart rate, WBC, blood glucose, creatinine, blood urea nitrogen, lactate, platelet count, prothrombin time (PT), arterial blood pH, $PaO_2$, $PaCO_2$, HCO3-, first-day oxygenation index, tidal volume, total respiratory rate, spontaneous respiratory rate, set respiratory rate, PEEP, plateau pressure, peak airway pressure, lung compliance, driving pressure, and mechanical power. After matching, 263 patients were included in the transpulmonary pressure monitoring group and 263 in the No-monitoring group. The balance between the matched groups was evaluated using the Standardized Mean Difference (SMD).

### Causal mediation analysis

Causal mediation analysis (CMA) is used to partition the total effect of a treatment into direct and indirect effects, helping to identify and quantify the mechanisms linking independent and dependent variables. When variable X influences variable Y through variable M, M is considered a mediator. The effect of X on Y, independent of M, is termed the average direct effect (ADE), while the combined effect of X on M and M on Y constitutes the average causal mediation effect (ACME). In this study, we hypothesized that a specific clinical indicator serves as a mediator variable. Transpulmonary driving pressure may induce changes in this indicator, which, in turn, influence mortality in ARDS patients. To further investigate the mediation effect, we analyzed APSIII, $PaCO_2$, peak airway pressure, plateau pressure, lung compliance, and driving pressure, given their differences between high and low transpulmonary driving pressure groups. ACME, ADE,

and the total effect derived from CMA were used to assess whether transpulmonary driving pressure affects mortality through specific mediators.

### Cluster analysis

Lasso regression was employed to identify variables associated with ICU mortality in patients receiving transpulmonary pressure monitoring. Right-skewed variables were logarithmically transformed to stabilize variance before standardization. The optimal number of clusters and K-means clustering distribution under resampling conditions were assessed using consensus clustering (CC), applying a sub-sampling ratio of 80% across 1,000 iterations and exploring cluster numbers (k) from 2 to 6. The best cluster number was determined based on the heatmap of the Consistency Matrix (CM), the Cumulative Distribution Function (CDF), and intra-cluster consistency scores. A consistency score approaching 1 indicated greater cluster stability. Once the optimal cluster was established, results were visualized using t-distributed stochastic neighbor embedding (t-SNE) and line graphs. The relationship between different clustering phenotypes and mortality was analyzed, with clinical characteristics expressed as percentages, mean ± standard deviation, or median with interquartile range. Group comparisons were conducted using the chi-square test, analysis of variance (ANOVA), and Wilcoxon's test. All statistical analyses were performed using R version 4.3.1.

## Results

### Patient enrollment and grouping

A total of 4,721 patients who met the inclusion criteria were included in the study. The patient selection flowchart is shown in Fig 1. Of these, 295 ARDS patients received targeted transpulmonary driving pressure (TPDP) management during mechanical ventilation (TPDP group), while 4,426 ARDS patients did not undergo targeted TPDP management (No-TPDP group). For patients in the TPDP group, the Survminer package was used to determine that the optimal cutoff value for transpulmonary driving pressure was 12.5 cmH$_2$O (S2 Fig). Based on this threshold, patients in the TPDP group were further categorized into a high TPDP group (n = 64, TPDP > 12.5 cmH$_2$O) and a low TPDP group (n = 231, TPDP ≤ 12.5 cmH$_2$O).

### ARDS severity-based analysis

The 4,721 ARDS patients were classified by severity according to the 2024 New Global Definition into mild, moderate, and severe categories. Patients with severe ARDS (n = 997) had the longest ICU stays and the highest 28-day, ICU, and hospital mortality rates (Table 1). Kaplan-Meier (K-M) survival curves showed that survival probabilities decreased as ARDS severity increased (Fig 2). These findings highlight the relationship between ARDS severity and patient outcomes, with severe ARDS patients exhibiting significantly worse prognoses than those in the mild and moderate groups.

### Impact of TPDP on survival by ARDS severity

Among the 295 ARDS patients in the TPDP group, the cohort was stratified into three severity groups based on P/F ratio: mild (200 mmHg < P/F ratio ≤ 300 mmHg), moderate (100 mmHg < P/F ratio ≤ 200 mmHg), and severe (P/F ratio ≤ 100 mmHg). No significant differences in mortality were observed across these severity groups (p > 0.05) (Table 2). However, in patients with moderate to severe ARDS, higher TPDP levels were significantly associated with increased 28-day, ICU, and hospital mortality rates (S2 Table). These findings suggest that the impact of TPDP on mortality may be more pronounced in patients with greater disease severity.

### Impact of target transpulmonary driving pressure values

Before propensity score matching (PSM), significant differences were observed between the TPDP and No-TPDP groups in age, BMI, severity scores, and key respiratory and laboratory parameters. After applying 1:1 PSM (n = 526, with 263

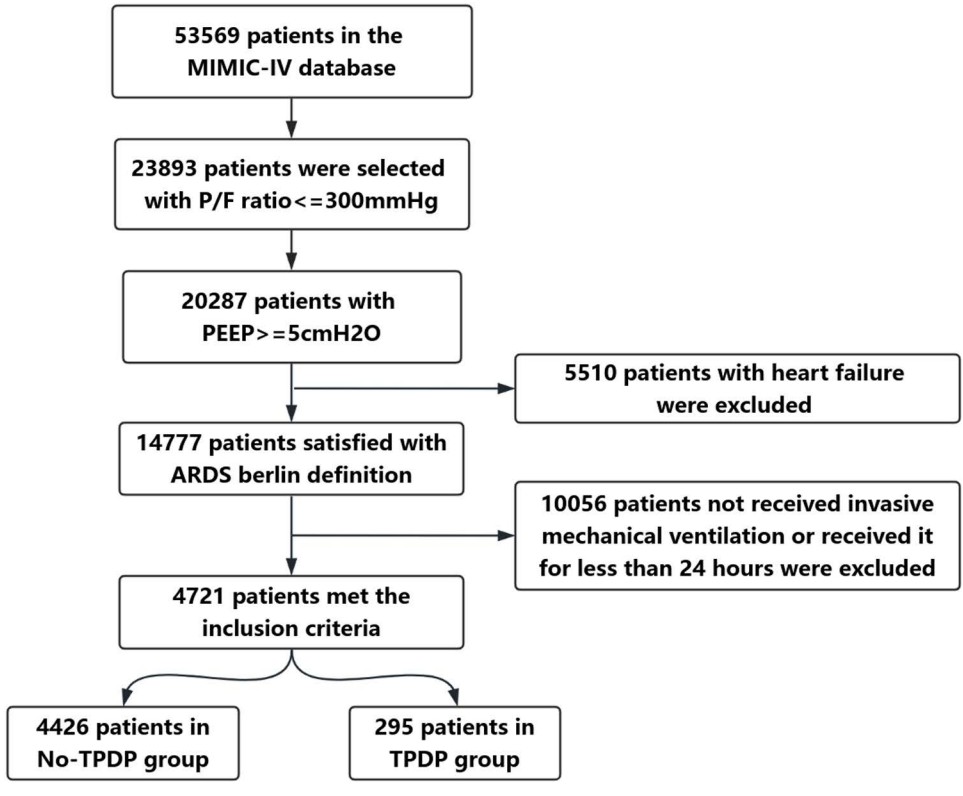

**Fig 1. Flow chart of ARDS patients enrollment and grouping.**

**Table 1. Prognosis based on ARDS severity categories.**

| Characteristic | N | Overall N=4,721 | P/F ratio≤100 mmHg N=997 | 100mmHg<P/F ratio≤200 mmHg N=1,910 | 200mmHg<P/F ratio≤300 mmHg N=1,814 | p-value |
|---|---|---|---|---|---|---|
| ICU staydays | 4,721 | 9.8±(9.1) | 10.4±(9.3) | 9.8±(9.4) | 9.3±(8.7) | <0.001 |
| Hospital staydays | 4,721 | 16.2±(15.6) | 15.9±(13.6) | 16.6±(17.2) | 16.1±(15.0) | >0.900 |
| 28-day survival day | 4,721 | 21.6±(10.2) | 20.4±(10.8) | 21.8±(10.2) | 22.2±(9.9) | <0.001 |
| 28-day mortality(%) | 4,721 | 1,468(31%) | 369(37%) | 583(31%) | 516(28%) | <0.001 |
| Hospital mortality(%) | 4,721 | 1,419(30%) | 358(36%) | 561(29%) | 500(28%) | <0.001 |
| ICU mortality(%) | 4,721 | 1,218(26%) | 313(31%) | 483(25%) | 422(23%) | <0.001 |

patients per group), these differences were balanced, and no significant variations in baseline characteristics remained between the groups (*p*>0.05) (Tables 3 and S3). Kaplan-Meier survival analysis indicated a higher survival probability in the TPDP group compared to the No-TPDP group (*p*=0.006) (Fig 3). Cox proportional hazards regression showed that target TPDP values were associated with a reduced risk of ICU mortality (HR 0.676, 95% CI 0.511–0.894, *p*=0.006). The 28-day survival analysis results showed no statistical significance and were provided in the supplementary materials (S3 Fig).

## Association between TPDP and mortality

In the TPDP group, each 1 cmH$_2$O increased in transpulmonary driving pressure was associated with a 1.177-fold increase in the risk of 28-day mortality, a 1.145-fold increase in ICU mortality risk, and a 1.144-fold increase in hospital

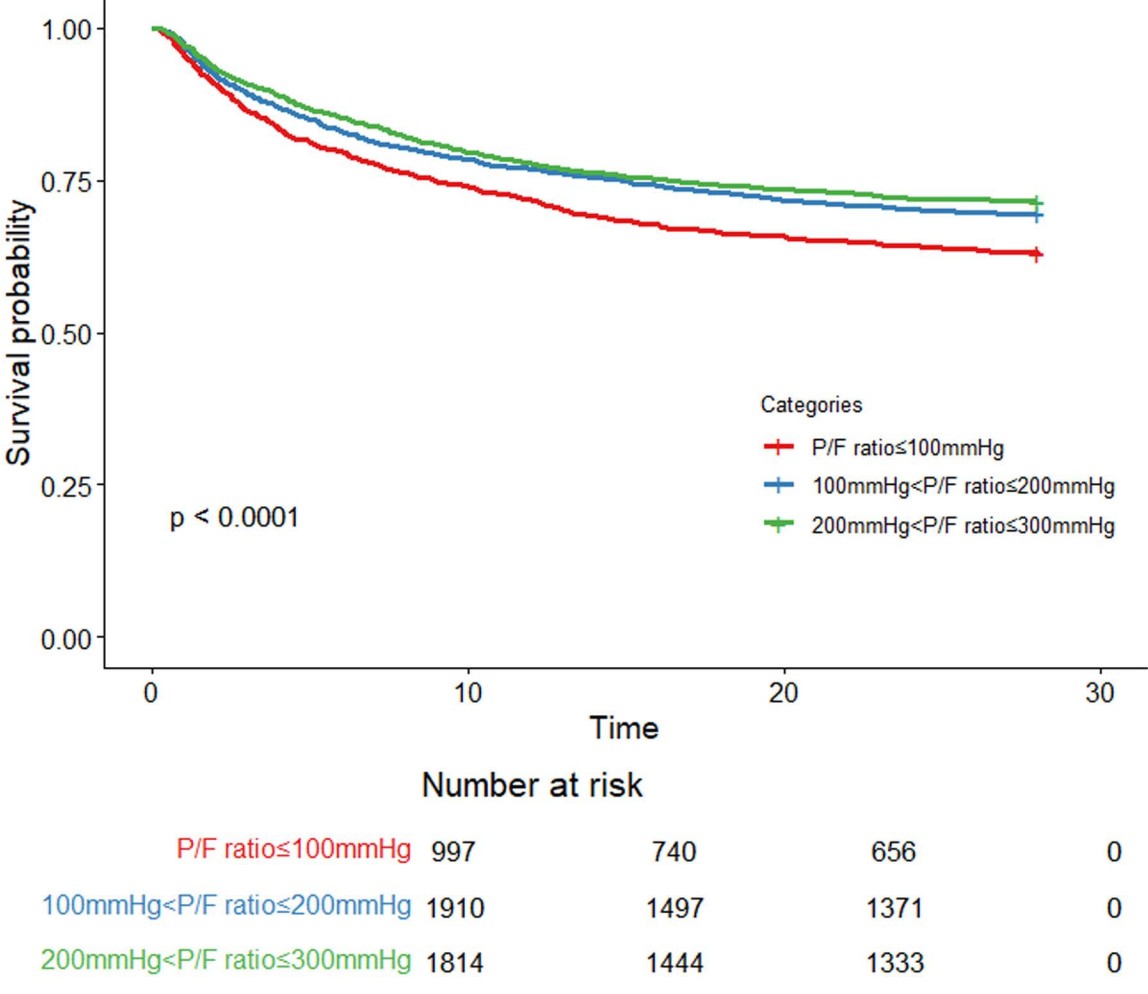

**Fig 2. Survival analysis based on ARDS severity.**

**Table 2. Prognosis based on ARDS severity categories in TPDP group.**

| Characteristic | N | Overall N=295 | P/F ratio≤ 100 mmHg N=159 | 100mmHg<P/F ratio≤ 200 mmHg N=107 | 200mmHg<P/F ratio≤ 300 mmHg N=29 | p-value |
|---|---|---|---|---|---|---|
| ICU staydays | 295 | 14.0±(12.3) | 13.6±(11.2) | 14.5±(14.2) | 15.0±(11.3) | 0.800 |
| Hospital staydays | 295 | 20.6±(19.4) | 19.6±(16.8) | 21.0±(22.0) | 24.8±(22.7) | 0.300 |
| 28-day survival day | 295 | 20.6±(10.7) | 20.5±(10.8) | 19.9±(10.9) | 24.2±(8.8) | 0.120 |
| 28-day mortality(%) | 295 | 107(36%) | 60(38%) | 42(39%) | 5(17%) | 0.078 |
| Hospital mortality(%) | 295 | 114(39%) | 64(40%) | 44(41%) | 6(21%) | 0.110 |
| ICU mortality(%) | 295 | 102(35%) | 57(36%) | 40(37%) | 5(17%) | 0.110 |

mortality risk. The ORs and 95% confidence intervals were 1.177 (1.101–1.258), 1.145 (1.073–1.222), and 1.144 (1.072–1.220), respectively, with *p*-values all below 0.05 (Fig 4). When stratified into high and low TPDP groups using a cutoff of 12.5 cmH$_2$O, both t-test and rank-sum test results showed that the high TPDP group had significantly higher APSIII,

 

**Table 3. The comparison between TPDP group and No-TPDP group.**

| | Original cohort | | P | Matched cohort | | P |
|---|---|---|---|---|---|---|
| | TPDP group | No-TPDP group | | TPDP group | No-TPDP group | |
| n | 295 | 4426 | | 263 | 263 | |
| Demographic characteristics | | | | | | |
| Age (median [IQR]) | 57.60 [43.85, 66.90] | 62.95 [52.10, 73.50] | <0.001 | 58.20 [45.25, 67.05] | 56.60 [43.25, 68.00] | 0.644 |
| BMI (median [IQR]) | 31.40 [26.65, 36.75] | 28.30 [25.00, 32.20] | <0.001 | 30.30 [26.25, 36.05] | 29.90 [25.20, 35.05] | 0.248 |
| The severity of illness | | | | | | |
| APSIII (median [IQR]) | 91.00 [66.00, 112.50] | 69.00 [50.00, 92.00] | <0.001 | 90.00 [65.00, 111.50] | 87.00 [62.50, 114.00] | 0.619 |
| SOFA score (median [IQR]) | 9.20 [7.25, 11.60] | 7.00 [4.80, 9.80] | <0.001 | 9.10 [7.20, 11.50] | 9.20 [6.50, 12.00] | 0.983 |
| Vital signs | | | | | | |
| ABPd (median [IQR]) | 58.10 [54.25, 62.30] | 57.60 [53.80, 62.27] | 0.307 | 58.10 [54.25, 62.25] | 57.80 [54.30, 62.45] | 0.977 |
| ABPs (median [IQR]) | 107.40 [102.10, 112.60] | 112.70 [106.20, 119.40] | <0.001 | 107.80 [102.05, 112.65] | 108.50 [101.95, 113.30] | 0.837 |
| Heart Rate (median [IQR]) | 94.00 [82.25, 106.00] | 86.40 [75.50, 98.60] | <0.001 | 92.90 [81.85, 104.70] | 94.40 [80.25, 105.40] | 0.988 |
| Blood gas analysis parameters | | | | | | |
| Arterial PH (median [IQR]) | 7.30 [7.30, 7.40] | 7.40 [7.30, 7.40] | <0.001 | 7.30 [7.30, 7.40] | 7.30 [7.30, 7.40] | 0.886 |
| PaO$_2$ (median [IQR]) | 92.60 [80.95, 108.75] | 112.90 [96.00, 132.48] | <0.001 | 93.10 [82.75, 110.70] | 93.50 [79.50, 109.20] | 0.455 |
| PaCO$_2$ (median [IQR]) | 43.80 [38.60, 49.20] | 39.80 [35.80, 44.30] | <0.001 | 43.38 (7.19) | 43.72 (7.47) | 0.601 |
| HCO3 (median [IQR]) | 21.00 [17.90, 24.50] | 22.00 [19.30, 25.00] | <0.001 | 21.00 [17.80, 24.55] | 21.00 [18.25, 24.50] | 0.526 |
| PFratio (median [IQR]) | 95.00 [68.80, 136.90] | 176.00 [116.00, 239.00] | <0.001 | 99.00 [71.70, 145.00] | 94.00 [71.00, 145.00] | 0.740 |
| Respiratory characteristics | | | | | | |
| Tidal Volume (mean (SD)) | 426.00 (78.48) | 469.92 (76.44) | <0.001 | 425.00 [365.05, 474.10] | 434.70 [382.00, 477.45] | 0.334 |
| Spontaneous Respiratory Rate (median [IQR]) | 0.00 [0.00, 1.00] | 0.90 [0.00, 5.00] | <0.001 | 0.00 [0.00, 1.10] | 0.00 [0.00, 1.10] | 0.226 |
| Total Respiratory Rate (median [IQR]) | 26.30 [22.70, 29.25] | 20.50 [17.50, 24.00] | <0.001 | 25.80 [22.30, 28.95] | 25.00 [21.90, 29.10] | 0.460 |
| Set Respiratory Rate (median [IQR]) | 25.30 [21.80, 28.00] | 18.50 [16.00, 22.00] | <0.001 | 24.70 [21.05, 27.50] | 24.00 [20.70, 27.35] | 0.377 |
| PEEP (median [IQR]) | 10.50 [9.65, 12.00] | 6.40 [5.00, 8.90] | <0.001 | 10.00 [9.35, 12.00] | 10.20 [9.00, 12.00] | 0.979 |
| Plateau Pressure (median [IQR]) | 26.90 [23.65, 29.60] | 19.40 [16.50, 22.70] | <0.001 | 26.30 [23.40, 29.00] | 26.40 [23.65, 29.00] | 0.708 |
| Peak Pressure (median [IQR]) | 30.70 [26.60, 33.95] | 22.90 [19.10, 26.70] | <0.001 | 30.00 [26.10, 33.25] | 30.40 [26.70, 33.45] | 0.526 |
| Lung compliance(median [IQR]) | 27.50 [21.85, 34.90] | 38.00 [30.40, 47.30] | <0.001 | 27.30 [21.60, 34.50] | 27.80 [22.55, 35.20] | 0.700 |
| Driving pressure (median [IQR]) | 15.30 [12.50, 18.65] | 12.30 [10.30, 14.80] | <0.001 | 15.55 (4.05) | 15.58 (3.72) | 0.946 |
| Mechanical power (median [IQR]) | 23.50 [18.55, 28.90] | 15.20 [11.20, 20.20] | <0.001 | 22.30 [17.90, 27.05] | 23.10 [16.90, 27.95] | 0.950 |

TPDP, Transpulmonary Driving Pressure; BMI, Body mass index; APSIII, Acute physiology score; SD, Standard deviation; IQR, Interquartile range.

PaCO$_2$, peak airway pressure, plateau pressure, and driving pressure compared to the low TPDP group ($p < 0.05$). Additionally, lung compliance was significantly lower in the high TPDP group ($p = 0.011$) (Tables 4 and S4).

## Causal mediation analysis

In ARDS patients with target transpulmonary driving pressure values, the impact of a TPDP > 12.5 cmH$_2$O on the 28-day mortality rate was 7.0% (95% CI 0.002–0.180; $p = 0.045$), and this effect was mediated through its impact on peak airway pressure (ACME: $p = 0.045$) (Fig 5).

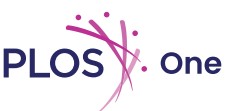

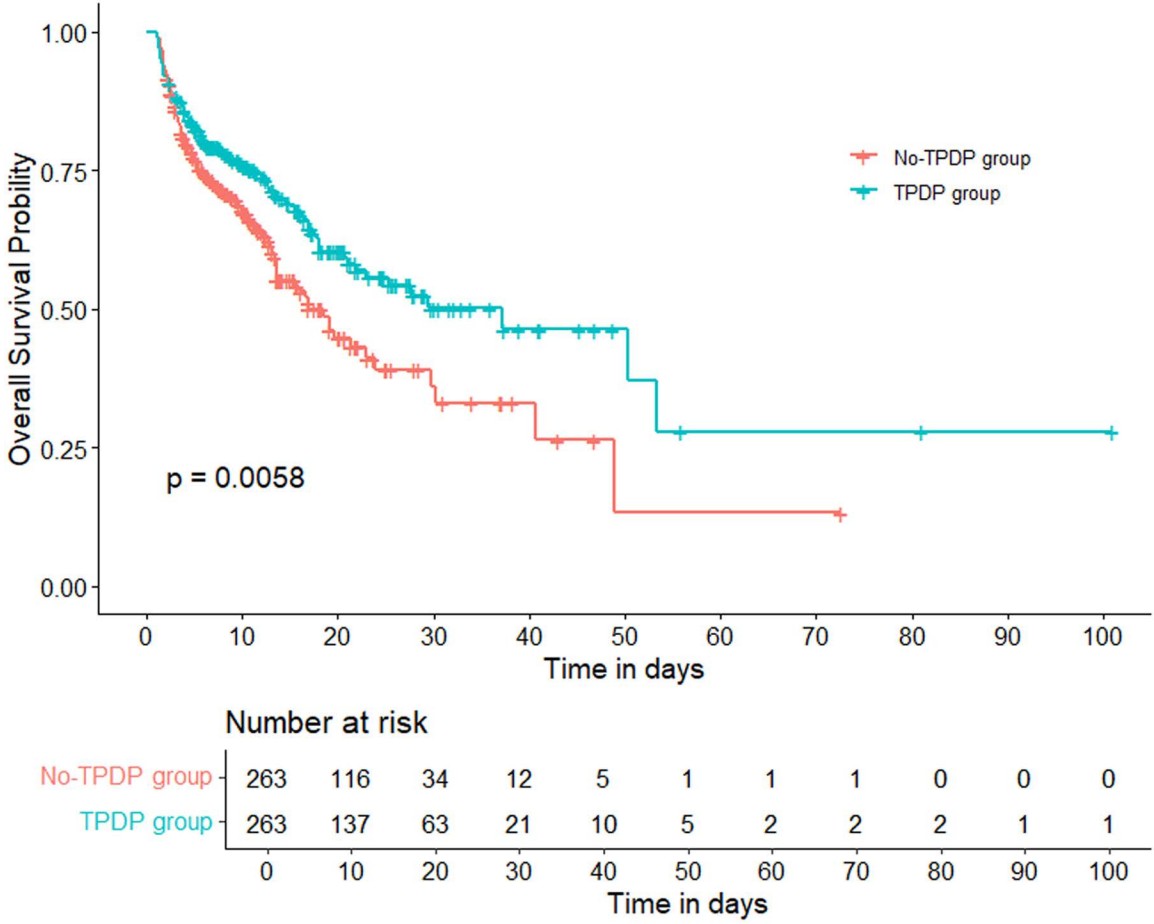

**Fig 3. Survival analysis of TPDP group and No-TPDP group after matching.**

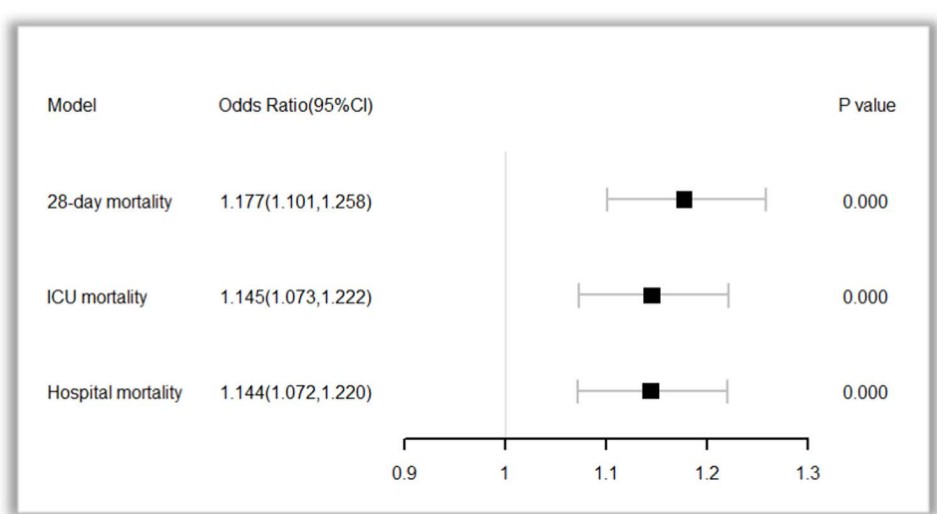

**Fig 4. The effect of transpulmonary driving pressure on mortality.**

**Table 4. The comparison between TPDP>12.5 cmH₂O group and TPDP ≤12.5 cmH₂O group.**

| | TPDP > 12.5 cmH₂O | TPDP ≤ 12.5 cmH₂O | P |
|---|---|---|---|
| n | 64 | 231 | |
| Blood gas analysis parameters | | | |
| Arterial PH (median [IQR]) | 7.30 [7.30, 7.32] | 7.30 [7.30, 7.40] | 0.121 |
| PaO₂ (median [IQR]) | 91.15 [80.12, 110.65] | 93.10 [81.85, 107.80] | 0.598 |
| PaCO₂ (mean (SD)) | 47.03 (8.01) | 43.14 (6.96) | 0.001 |
| HCO3 (median [IQR]) | 22.15 [18.78, 25.35] | 20.80 [17.80, 23.90] | 0.096 |
| PFratio (median [IQR]) | 91.00 [63.00, 129.98] | 95.20 [70.35, 139.55] | 0.466 |
| Respiratory characteristics | | | |
| Tidal Volume (mean (SD)) | 417.57 (81.76) | 428.33 (77.56) | 0.333 |
| Spontaneous Respiratory Rate (median [IQR]) | 0.00 [0.00, 0.30] | 0.00 [0.00, 1.10] | 0.223 |
| Total Respiratory Rate (median [IQR]) | 27.45 [22.75, 30.72] | 26.20 [22.70, 29.00] | 0.201 |
| Set Respiratory Rate (median [IQR]) | 24.75 [21.95, 28.72] | 25.30 [21.70, 28.00] | 0.817 |
| PEEP (median [IQR]) | 10.15 [9.75, 12.00] | 10.70 [9.65, 12.00] | 0.982 |
| Plateau Pressure (median [IQR]) | 28.30 [25.32, 31.30] | 26.50 [23.30, 29.20] | 0.001 |
| Peak Pressure (median [IQR]) | 32.05 (5.06) | 29.89 (5.09) | 0.003 |
| Lung compliance(median [IQR]) | 24.75 [19.92, 31.75] | 28.20 [22.65, 36.15] | 0.011 |
| Driving pressure (mean (SD)) | 16.66 (3.90) | 15.09 (4.09) | 0.006 |
| Mechanical power (median [IQR]) | 24.45 [19.38, 30.72] | 23.40 [18.30, 28.55] | 0.243 |
| Outcomes | | | |
| Mechanical ventilation hour (median [IQR]) | 138.50 [64.78, 244.32] | 118.80 [73.35, 184.80] | 0.238 |
| ICU staydays (median [IQR]) | 8.95 [4.57, 17.18] | 10.70 [6.10, 19.85] | 0.195 |
| Hospital staydays(median [IQR]) | 13.45 [4.40, 17.83] | 17.70 [9.05, 27.55] | 0.005 |
| 28-day survival day(median [IQR]) | 15.20 [3.70, 28.00] | 28.00 [17.00, 28.00] | <0.001 |
| 28-day mortality (%) | 41 (64.06) | 66 (28.57) | <0.001 |
| Hospital mortality(%) | 40 (62.50) | 74 (32.03) | <0.001 |
| ICU mortality (%) | 38 (59.38) | 64 (27.71) | <0.001 |

TPDP, Transpulmonary Driving Pressure; BMI, Body mass index; APSIII, Acute physiology score; SD, Standard deviation; IQR, Interquartile range.

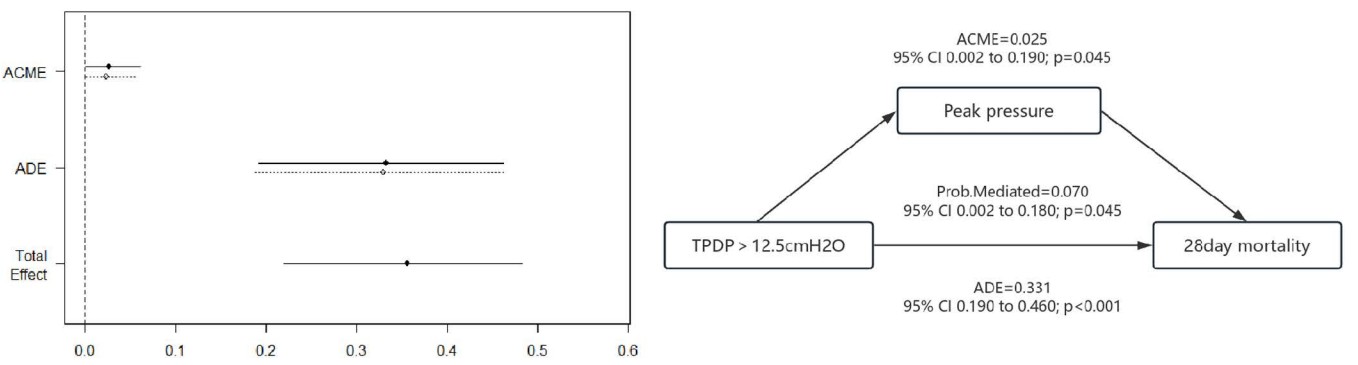

**Fig 5. Causal mediation analysis.**

**Table 5. The relationship between high and low TPDP group in difference phenotypes.**

| Variable | Phenotype | Hazard Ratio (HR) | Lower Bound 95% CI | Upper Bound 95% CI | p-value |
|---|---|---|---|---|---|
| 28d mortality | Phenotype-I | 5.267 | 3.016 | 9.198 | <0.001 |
| | Phenotype-II | 3.213 | 1.262 | 8.182 | 0.010 |
| | Phenotype-III | 0.971 | 0.443 | 2.131 | 0.940 |
| Hospital mortality | Phenotype-I | 4.446 | 2.582 | 7.655 | <0.001 |
| | Phenotype-II | 2.628 | 1.043 | 6.622 | 0.033 |
| | Phenotype-III | 0.949 | 0.433 | 2.080 | 0.900 |

## Cluster analysis

Using the glmnet package, the optimal λ value for model performance was determined to be 0.023, which provided the best predictive accuracy while minimizing the number of selected variables (S5 Fig). Fourteen predictors were ultimately retained: TPPinsp, TPDP, age, BMI, heart rate, WBC, BUN, PT, platelet count, Lac, arterial pH, P/F ratio, tidal volume, and peak pressure. Consensus clustering analysis identified three major sample clusters. The cumulative distribution function (CDF) graph illustrated the distribution of CDF across different cluster numbers, with the Δarea graph indicating the most pronounced changes in the area under the CDF curve occurring between k = 2 and k = 4 (S6 Fig). The heatmap of the consensus matrix revealed clear separation at k = 2 and k = 3 (S7 Fig). Stability assessment of clustering results showed that average cluster consistency scores exceeded 0.8 for both k = 2 and k = 3 (S8 Fig), leading to the conclusion that three clusters provided the optimal classification.

Consensus clustering analysis thus identified three distinct clinical phenotypes among ARDS patients, visualized using t-SNE plots (S9 Fig). The characteristics of the three ARDS phenotypes are summarized in S5 Table. Phenotype-I (n = 138) exhibited the higher mechanical power (median 24.85 J/min [IQR 19.02–30.50]) alongside moderate lung compliance (28.65 mL/cmH$_2$O [24.33–34.68]). Phenotype-II (n = 62) demonstrated elevated spontaneous respiratory rates (median 0.75 breaths/min [IQR 0.00–2.60]) and the highest lung compliance (36.95 mL/cmH$_2$O [31.40–46.88]), suggesting preserved pulmonary elasticity. Phenotype-III (n = 95) was characterized by markedly reduced tidal volumes (346.60 mL [320.65–364.35]) and the worst lung compliance (20.80 mL/cmH$_2$O [17.70–25.95]), indicative of severe restrictive pathophysiology. Significant inter-phenotype differences were observed in tidal volume, respiratory rates, mechanical power, and compliance (all p < 0.001), highlighting distinct ventilatory management requirements and potential prognostic implications.

## Heterogeneity of transpulmonary driving pressure effects in different phenotypes

The impact of transpulmonary driving pressure on patient prognosis was analyzed across the three phenotypes. Each phenotype was further categorized into high and low TPDP groups using 12.5 cmH$_2$O as the cutoff. The effect of transpulmonary driving pressure varied among different phenotypes. Log-rank test results showed that in Phenotype-I and Phenotype-II, patients with high TPDP had significantly higher hospital and 28-day mortality compared to those in the low TPDP group. However, in Phenotype-III, transpulmonary driving pressure did not significantly affect mortality (Table 5).

## Discussion

### Phenotype-specific mechanisms and clinical implications of TPDP in ARDS

Our study demonstrates that elevated transpulmonary driving pressure shows significant association with increased mortality in acute respiratory distress syndrome patients. Recent investigations have further underscored the critical prognostic implications the TPDP in ARDS management. Placenti et al [22] established that TPDP more reliably reflects lung distension pressures compared to airway driving pressure, representing the actual transpulmonary pressure gradient

without the distending pressure generated by the chest wall or esophageal driving pressure. In a study by Baedorf Kassis et al [23], treatment strategies leading to decreased transpulmonary driving pressure may be associated with improved 28 day mortality. This finding is consistent with the 2024 study by Sun et al, where TPDP should be used as a key monitoring indicator for lung protective ventilation [24].

Our study further revealed that when TPDP exceeds 12.5 cmH$_2$O, patient mortality significantly increases. This threshold closely aligns with the 'safety window' (10–12 cmH$_2$O) proposed by Placenti et al [22], suggesting that 12.5 cmH$_2$O may represent the critical threshold for triggering ventilator-induced lung injury (VILI). This relationship may be partially mediated through higher peak airway pressure. The detrimental clinical impacts of elevated peak airway pressures have been well documented. Experimental evidence from a rat mechanical ventilation model developed by Katira et al [25] revealed that excessive peak airway pressure induce right ventricular overload through increased pulmonary vascular capacitance and perfusion abnormalities. It is suggested that control of peak airway pressure can interrupt the vicious cycle of "high TPDP→ alveolar hyperinflation → pulmonary circulation injury". Our causal mediation analysis substantiates the hypothesis that therapeutic strategies targeting TPDP reduction could mitigate both peak airway pressure elevation and subsequent pulmonary damage. This mechanistic understanding emphasizes the potential clinical utility of TPDP-guided ventilation strategies in ARDS management.

The differential effects of transpulmonary driving pressure (TPDP) across ARDS phenotypes arise from distinct pathophysiological mechanisms and heterogeneous lung mechanical properties. In Phenotype I (High Mechanical Power with Moderate Lung Compliance), the preserved lung compliance indicates partial alveolar recruitability. However, excessive mechanical power potentiates VILI via bioenergetic stress-induced tissue damage and inflammatory cascades [26], where elevated TPDP potentiates ventilator-induced lung injury through biomechanical transduction pathways. Phenotype II (High Spontaneous Effort with Better Compliance) demonstrates paradoxically increased lung compliance combined with vigorous spontaneous breathing effort, creating conditions favorable for pendelluft phenomenon [27]. In this context, TPDP elevation amplifies cyclic alveolar collapse-reopening cycles in dependent lung zones, thereby exacerbating atelectrauma through shear stress mechanisms [28]. Phenotype III (Low Tidal Volume with Reduced Compliance), characterized by low tidal volume and markedly diminished compliance, exhibits limited TPDP transmission due to mechanical stress dissipation through No-recruitable lung parenchyma. This stress-shielding effect explains the absence of significant mortality correlation despite TPDP fluctuations. This mechanistic stratification underscores the clinical imperative for phenotype-driven ventilation strategies. Our observations harmonize with contemporary evidence from Liu et al [29], reinforcing the paradigm shift toward precision medicine in ARDS management.

## The berlin criteria vs. the new global definition of ARDS

We selected the Berlin criteria due to its established integration with mechanical ventilation parameters. The 2024 Global ARDS Definition includes No-intubated patients and lacks standardized ventilation settings [30], whereas the Berlin criteria require PaO$_2$/FiO$_2$ assessment under PEEP ≥5 cmH$_2$O, ensuring consistency. Additionally, moderate-to-severe ARDS classification (PaO$_2$/FiO$_2$ ≤ 200 mmHg) is widely validated for prognostic assessment, while the broader mild ARDS category in the new definition remains unverified in ventilated patients, increasing heterogeneity and potentially diluting effect size. Our findings reinforce this choice, as TPDP >12.5 cmH$_2$O correlated most strongly with mortality in moderate-to-severe ARDS, a link consistent with ventilator-induced lung injury mechanisms. A recent MIMIC-IV study by Ye et al [31] continue to rely on the Berlin criteria, underscoring its relevance in mechanically ventilated patients.

## Strengths and limitations of database utilization

MIMIC-IV enables large-scale retrospective ARDS analysis, enhancing generalizability. Its detailed respiratory and physiological data facilitate an in-depth evaluation of TPDP. However, it lacks granular details on ARDS onset, therapeutic interventions, and individualized management, precluding direct causality assessment. Mediation analysis provides inferential



insights but requires validation in prospective studies. Additionally, the absence of bedside-level details limits direct clinical applicability.

### Internal and external validity considerations

The study's internal validity is reinforced by rigorous statistical adjustments, including propensity score matching and mediation analysis, aimed at reducing confounding effects. However, given the retrospective design, residual confounding cannot be entirely ruled out due to potential unmeasured variables. External validity is supported by the multi-institutional nature of the MIMIC-IV database, allowing broader applicability to healthcare centers with similar resources and clinical practices. Noetheless, findings may not be generalizable to institutions with different patient populations, ventilatory management protocols, or resource availability, highlighting the need for validation in diverse clinical environments.

### Study limitations and future directions

Several limitations should be considered when interpreting our findings. First, it should be noted that our findings are derived from retrospective observational data, and the evidence supporting the role of transpulmonary driving pressure in influencing mortality is indirect. Causality cannot be inferred from these results. Therefore, proof of concept studies using animal models and prospective interventional trials are needed to validate the physiological mechanisms and causal links suggested by our findings. Second, the lack of standardized TPDP measurement protocols within MIMIC-IV introduces potential variability in data accuracy and reproducibility. Third, while our phenotype-based subgroup analysis provides novel insights into differential responses to TPDP, further validation through prospective cohort studies and mechanistic investigations is needed. Future research should prioritize controlled clinical trials to determine the optimal TPDP threshold, clarify its interaction with mechanical power, and evaluate the impact of phenotype-specific ventilatory strategies on ARDS outcomes.

## Conclusions

Our study suggests that TPDP >12.5 cmH$_2$O may be associated with an increased risk of mortality in ARDS patients, particularly those with moderate to severe ARDS. Incorporating target TPDP values into routine clinical practice might enhance the precision of mechanical ventilation strategies, potentially reducing ventilator-induced lung injury and improving patient outcomes. However, recognizing the different ARDS phenotypes and tailoring ventilation strategies based on these profiles could further contribute to better patient prognosis. Future research should aim to validate these findings, develop phenotype-specific management protocols, and explore the mechanisms underlying the varied responses to TPDP in ARDS patients. This will help provide a more comprehensive understanding of the role of TPDP in the management of ARDS.

## Supporting information

**S1 Fig. The distribution of missing data.**
(TIF)

**S2 Fig. The selection of optimal cut-off value of transpulmonary driving pressure.** TPDP, Transpulmonary driving pressure; grps, groups.
(TIF)

**S3 Fig. 28-day survival analysis of TPDP group and No-TPDP group after matching.**
(TIF)

**S4 Fig. Changes in standardized mean difference before and after matching.** PEEP, Positive end expiratory pressure; PFratio, P/F ratio of the first day; APSIII, Acute physiology score; ABPs, Systolic arterial pressure; BMI, Body mass



index; Lac, Lactate level; PT, Prothrombin time; BUN, Urea nitrogen; WBC, White blood cell; ABPd, Diastolic blood pressure.
(TIF)

**S5 Fig. The trend of mean squared error with respect to the λ parameter in the LASSO regression model.**
(TIF)

**S6 Fig. (A) Consensus cumulative distribution function (CDF) distributions with different cluster numbers k from 2 to 6. (B) Delta area of consensus CDF for number of clusters cluster numbers k from 2 to 6.**
(TIF)

**S7 Fig. The consensus matrix heat map of consensus cluster analysis illustrates the agreement values for each cluster (k) in patients with ARDS, represented on a white to blue color scale.**
(TIF)

**S8 Fig. The bar plot of consensus cluster analysis.**
(TIF)

**S9 Fig. T-SNE visualization of phenotype assignments by consensus clustering.**
(TIF)

**S1 Table. Normality test of variables between groups.**
(DOCX)

**S2 Table. The association between TPDP levels and mortality.**
(DOCX)

**S3 Table. The comparison of laboratory test and outcomes between TPDP group and No-TPDP group.**
(DOCX)

**S4 Table. The comparison of Demographic characteristics, The severity of illness, Vital signs, and Laboratory test between TPDP > 12.5 cm $H_2O$ group and TPDP ≤12.5 cm$H_2O$ group.**
(DOCX)

**S5 Table. Characteristics of the three phenotypes.**
(DOCX)

## Acknowledgments

The authors extend their gratitude to the researchers who established and maintained the MIMIC-IV database. The opinions expressed in this study are solely those of the authors and do not represent the views of any affiliated third parties.

## Author contributions

**Conceptualization:** Mingxing Fang.

**Data curation:** Na Liu, Qi Zhang.

**Formal analysis:** Na Liu, Qi Zhang, Huiyong Wang, Renshuang Ding.

**Funding acquisition:** Mingxing Fang.

**Investigation:** Na Liu, Qi Zhang, Huiyong Wang, Renshuang Ding.

**Methodology:** Na Liu, Qi Zhang.



**Project administration:** Mingxing Fang.

**Writing – original draft:** Na Liu, Qi Zhang.

**Writing – review & editing:** Xiaoyong Geng, Li Wang, Zhiyong Wang.

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
