## [Decision Letter · Decision Letter 0]

Dear Dr. Fang,

Thank you for submitting your manuscript to PLOS ONE. After careful consideration, we feel that it has merit but does not fully meet PLOS ONE’s publication criteria as it currently stands. Therefore, we invite you to submit a revised version of the manuscript that addresses the points raised during the review process.

We look forward to receiving your revised manuscript.

Kind regards,

Gurmeet Singh, M.D., Ph.D.,

Academic Editor

PLOS ONE

2. Thank you for stating the following financial disclosure: [FMX was supported by fundings of the government of Hebei Province, China funded the provincial Medical Excellence Program and high-level talents funding Program of Hebei Province, China (grant number A20203005).]. Please state what role the funders took in the study. If the funders had no role, please state: "The funders had no role in study design, data collection and analysis, decision to publish, or preparation of the manuscript." If this statement is not correct you must amend it as needed. Please include this amended Role of Funder statement in your cover letter; we will change the online submission form on your behalf.

3. Thank you for uploading your study's underlying data set. Unfortunately, the repository you have noted in your Data Availability statement does not qualify as an acceptable data repository according to PLOS's standards. At this time, please upload the minimal data set necessary to replicate your study's findings to a stable, public repository (such as figshare or Dryad) and provide us with the relevant URLs, DOIs, or accession numbers that may be used to access these data. For a list of recommended repositories and additional information on PLOS standards for data deposition, please see https://journals.plos.org/plosone/s/recommended-repositories.

Additional Editor Comments (if provided):

Reviewers' comments:

Reviewer's Responses to Questions

**Comments to the Author**

1. Is the manuscript technically sound, and do the data support the conclusions?

Reviewer #1: Yes

2. Has the statistical analysis been performed appropriately and rigorously?

Reviewer #1: Yes

3. Have the authors made all data underlying the findings in their manuscript fully available?

Reviewer #1: Yes

4. Is the manuscript presented in an intelligible fashion and written in standard English?

Reviewer #1: Yes

Reviewer #1: The study by Dr. Liu et al evaluated the effects of transpulmonary driving pressures on ARDS outcome. The relationship between ARDS outcome and transpulmonary pressures during mechanical ventilation is not novel and has been investigated as transpulmonary pressures, ventilation driving pressures and even transpulmonary driving pressure. All consistently imply that intrapulmonary pressure control during mechanical ventilation leads to better outcomes. In this study, the authors grouped patients into new phenotypes; some that are more affected by transpulmonary driving pressures. The latter is interesting information but needs further testing in randomized controlled studies.

There has been considerable research into patients with ARDS in order to improve outcomes. It is now well established that ARDS is a complex collection of pathologies that differ in predictive and prognostic variables. Therefore, patient selection for clinical investigation is central to successful study design.

Patient selection benefits from prognostic enrichment; enrolling patients with a high probability of an actionable outcome. This will increase the power to detect the desired outcome of the study. The best example of successful prognostic enrichment in ARDS is the PROSEVA trial that studied the effects of prone positioning. Enrolling patients with moderate-to-severe ARDS (those with PaO2/FiO2 less than 150 mmHg) showed a mortality benefit. This observation might have been lost without prognostic enrichment. Studies that do not enrich their enrollment for severity of disease or an actionable outcome become inadvertently underpowered. This approach has to be balanced against limitations in the ability to generalize the data.

There is no amount of statistical matching that can replace strategic enrollment. This study would have benefited from a more targeted approach. Similarly, various ICU severity of illness scores were tested but they are not specific for ARDS and therefore do not help normalize or quantify the severity of illness in a study group. The authors may want to rethink their enrollment strategy.

Specific Points:

The authors suggest that ARDS outcomes are influenced by the measurement of the transpulmonary driving pressure. This is erroneous. Rather, it would be the choice of target pressure values that are the important variables. The authors explain this later in their study, but the title and text should be changed to reflect this.

Both the 2016, Berlin and 2024, New Global Definition of ARDS include severity categories since these are strong indicators of outcomes. The patients included in this study should have been analyzed within these categories as the severity grouping are prognostic by themselves.

New Global Definition of ARDS 2024

• Mild: PaO2/FiO2 >200 mm Hg, but ≤300 mm Hg

• Moderate: PaO2/FiO2 >100 mm Hg, but ≤ 200 mm Hg

• Severe: PaO2/FiO2 ≤ 100 mm Hg

Please include transpulmonary driving pressures in the Methods under respiratory data extracted from the database. The Methods only describe extracting “driving pressures” which are usually calculated using the equation ΔP = Pplat – PEEP and differ from transpulmonary driving pressures. This could be confusing to the reader.

The authors should explain how transpulmonary driving pressures were measured.

The calculation of TPDP is well described; using Esophageal pressure (Pes) as a surrogate for Ppl or, the PEEP step up approach. If the latter procedure was used, it is necessary to explain if a one or two PEEP step up was used and if a best fit lung P/V curve was constructed. It is important to specify the method as The Gattinoni group has shown that during a PEEP trial the overall ΔPTP/ΔPAW was 0.71, but ranged from 0.36 at PEEP 5 cmH2O in patients with extrapulmonary ARDS, to 0.98 at PEEP 15 cmH2O in patients with pulmonary ARDS.

The Results can be simplified. Please describe your findings and use Tables/Figures as a reference point. There is too much data in each Table for a reader to evaluate. Can you reduce the Table into smaller yet logical collections of information?

The Discussion could be markedly shortened and needs reorganization. Right now, the Discussion reads like pieces of a review paper. The Discussion is literary space used to interpret your findings and put them in context of current practice. Please use the first paragraph to summarize your findings and explain how the TPDP findings from your study will influence clinical practice. Are you recommending all ICU facilities record TPDP? Or are you suggesting that they should be used in some of your phenotypes? Please briefly mention how to validate your phenotypes and if they differ from the three recognized categories of ARDS severity of illness.

Please explain how the data was optimized for analysis, e.g., propensity matching. Also explain any issues that the propensity matching does not address and how you resolved these. The information about databases can be reduced into one or two sentences and should pertain to the effects on your data analysis including how you resolved any shortcomings. Studies on transpulmonary pressures (DP, TPP and TPDP) have already been described in the peer reviewed literature. Take time to explain what makes this study different from previous studies and whether the existing literature supports the authors’ findings and clinical recommendations? Also explain if your study findings can be generalized to other sites of practice. If they cannot, then explain why they are still clinically important for the reader.

**Do you want your identity to be public for this peer review?** For information about this choice, including consent withdrawal, please see our Privacy Policy

Reviewer #1: **Yes: ** Mercedes Susan Mandell

---

## [Author Response · Author response to Decision Letter 1]

30 Dec 2024

Dear Editor and Reviewer,

We appreciate the insightful feedback on our manuscript and have revised the paper accordingly. Below, we outline our responses and corresponding changes:

1. Title and Content Revision:

We have revised the title and introductory sections to clarify the study's focus on target transpulmonary driving pressure (TPDP) values rather than the measurement of TPDP alone. This change aligns the title and text with the study's emphasis on target pressure values as the key influencing factors for ARDS outcomes (Page 1 lines1-4).

2. Incorporating ARDS Severity-Based Analysis:

Based on the reviewer’s recommendation, we have now included an analysis of ARDS severity categories according to both the Berlin Definition and the 2024 New Global Definition (mild, moderate, and severe) as indicators of patient prognosis. This stratified analysis is now reflected in the Results section (Page 6-7, lines 228-248), providing a clearer perspective on the impact of target TPDP values within distinct ARDS severity groups.

3. Clarifying TPDP Measurement in the Methods Section:

We expanded the Methods section to clarify the measurement approach for TPDP, differentiating it from conventional driving pressure. We specify that TDP was measured as the difference between end-inspiratory and end-expiratory transpulmonary pressures (Page 4, lines 140-144).

4. Simplifying the Results Section:

We streamlined the Results by focusing on essential findings and referring readers to the Tables and Figures for detailed data. We also reorganized extensive tables into smaller, logically grouped collections to improve readability and make the data easier to interpret (Page 7-9, lines 249-280).

5. Reorganizing and Condensing the Discussion:

The Discussion section was restructured to begin with a summary of key findings and the clinical implications of TPDP. We addressed whether we recommend TPDP monitoring for all ICU settings or selectively for specific ARDS phenotypes, and briefly mentioned the validation plans for the phenotypes identified. We also rephrased parts of the Discussion to interpret our findings within the context of current clinical practice, focusing on the study’s direct relevance to ICU management (Page 11-12, lines 323-383).

6. Clarifying Data Optimization and Analysis:

In response to the reviewer’s comments, we added details on the data optimization process, including the propensity score matching (PSM) approach and any limitations that remain unaddressed by PSM. We condensed the discussion of database-related details to focus on data analysis and methods for addressing limitations (Page 12, lines 367-374).

7. Comparison with Existing Literature:

We highlighted how this study contributes to the body of research on TPDP and TPDP, explaining how our findings align with or differ from prior studies. (Page 12, lines 375-383).

These revisions align our study more closely with the reviewer’s recommendations, enhancing both clarity and clinical relevance. Thank you for the valuable suggestions, and we look forward to your feedback on the revised manuscript.

---

## [Decision Letter · Decision Letter 1]

Dear Dr. Fang,

Thank you for submitting your manuscript to PLOS ONE. After careful consideration, we feel that it has merit but does not fully meet PLOS ONE’s publication criteria as it currently stands. Therefore, we invite you to submit a revised version of the manuscript that addresses the points raised during the review process.

Dear Dr Fang

<h3>The Reviewers has concluded that your manuscript entitled The Effect of Target Transpulmonary Driving Pressure Values on Mortality in ARDS Patients: A Retrospective Study Based on the MIMIC-IV Database needs major revision based on several points of considerations.</h3>

The comments related to the manuscript can be checked from the attached files.

We appreciate the fact that you have considered Plos One for the publication of your work. 

We look forward to receiving your revised manuscript.

Kind regards,

Gurmeet Singh, M.D., Ph.D.,

Academic Editor

PLOS ONE

Additional Editor Comments :

Major Revision

Reviewers' comments:

Reviewer's Responses to Questions

**Comments to the Author**

Reviewer #1: (No Response)

Reviewer #2: (No Response)

2. Is the manuscript technically sound, and do the data support the conclusions?

Reviewer #1: Partly

Reviewer #2: Partly

3. Has the statistical analysis been performed appropriately and rigorously?

Reviewer #1: Yes

Reviewer #2: I Don't Know

4. Have the authors made all data underlying the findings in their manuscript fully available?

Reviewer #1: Yes

Reviewer #2: Yes

5. Is the manuscript presented in an intelligible fashion and written in standard English?

Reviewer #1: Yes

Reviewer #2: No

Reviewer #1: The authors improved the manuscript. The Discussion still falls short of the bare minimum needed for publication. I provided a detailed roadmap for the authors to follow when constructing their Discussion.

Reviewer #2: 1. I thank the publisher for sending me this manuscript for review and apologize for the delay in performing the same to both the publisher and the authors.

2. In this paper, which I see in R1, the authors aimed to analyze the impact of transpulmonary on mortality in patients with ARDS and explored the prognostic implications of different levels of transpulmonary pressure all this by extrapolating and analyzing data from the MIMIC IV database

3. Interesting work, involving a large amount of data and a lot of statistical work associated with it.

Adequate developmental work until the discussion then during the discussion the authors lose the structure of the work and do not develop this section at all.

4. One thing absolutely to be checked further in the work is the grammar and spelling of the work, which is often deficient and the sentence structure at times very complex and unintelligible. I recommend a native language proofreading or by an appropriate service appointed for this.

Below are timely comments on the work

Abstract

5. Abstract to be restructured and recreated in relation to comments below. I recommend in the abstract to eliminate all abbreviations which are never recommended in this part of the work as they only weigh down the work and the section, and make it complicated to read.

Introduction

6. I find the introduction adequate and well developed and argued. Some points to be clarified. Especially by including appropriate references to the statements made by the authors.

7. Line 92-96 to insert a reference to this statement

8. Line 99-103 Insert an appropriate reference here as well.

Materials and methods

Selection of patients

9. I do not understand why it was chosen by the authors to use the Berlin definition and not the new broader definition for ARDS criteria ( doi: 10.1164/rccm.202303-0558WS. ). To justify this choice.

Statistical analysis.

10. Regarding the statistical analysis of the work I go into the structural aspect of the work. All development points should be described under the big chapter of statistical analysis, possibly the other points will result subchapters of the same.

11. To be properly restructured this section. The main section is the statistical analysis of the data.

Discussion

12. Line 331-381 This part of the discussion is a repetition, in some places, almost without difference, of the background described above. The discussion should “discuss” as the word says, the results obtained, all comparing them with the present literature and evaluating, point by point, whether in favor or different, looking for reasons, valid, supported by as much literature. Instead, in this part, the authors just rewrote something that was unnecessary. This part is all to be removed.

13. I advise the authors to restructure this part better, to really compare themselves with the literature, maybe even with some recent literature trying to properly evaluate their results.

14. Perhaps an idea would be to take each result point by point and develop a discussion on them, point by point, with the appropriate literature to accompany the results and the discussion.

Conclusions

15. I would advise the authors to be softer and softer with the conclusions, it is important, even considering the limitations of the study, to try to use the conditional and carefully evaluate the words in the conclusions of the study. To be corrected.

Figure 1

16. I advise the authors to avoid color and focus mainly on the accuracy of the concepts they want to get across, the first BOX is not clear, which patients were identified by MIMIC?

17. In the BOX where the authors state the patients meeting the Berlin criteria for ARDS is the number to be entered first, since the others were all structured this way, as were the other two groups, either all with n in parentheses, or all with numbers at the beginning.

18. To be restructured properly and logically in the figure.

Figure 2

19. I don't find this figure useful, in fact, it seems to me almost a given that if the patient has a P/F <100 they are more likely to die, or less likely to survive, depending on how we want to look at it, than a patient with a PF between 200 and 300, what is the reason for including this figure?

Figure 3

20. Interesting KM at 90 days on survival, it would also be interesting to have one, equal to 28 or 30 days, to better understand the difference, maybe trying to “zoom in” excuse the term, removing the unnecessary part of the graph. I don't know if I explained myself. In the end, it turned out a figure that the authors took into consideration, the 28-day mortality/survival if I remember correctly.

**Do you want your identity to be public for this peer review?** For information about this choice, including consent withdrawal, please see our Privacy Policy

Reviewer #1: **Yes: ** Mercedes Susan Mandell

Reviewer #2: No

---

## [Author Response · Author response to Decision Letter 2]

22 Mar 2025

Dear Editors and Reviewers:

Thank you for your letter and for the Reviewers’ comments concerning our manuscript entitled “The Effect of Target Transpulmonary Driving Pressure Values on Mortality in ARDS Patients: A Retrospective Study Based on the MIMIC-IV Database”. (Submission ID PONE-D-24-38166R1). Those comments are all valuable and very helpful for revising and improving our paper, as well as the important guiding significance to our researches. We have studied comments carefully and have made correction which we hope meet with approval. Revised portion are marked in red in the paper. The main corrections in the paper and the responds to the reviewer's comments are as following:

Responses to the Reviewer 1’s comments:

1.Reviewer’ s comment: The reviewer pointed out a limitation related to the assumption that patients with measured transpulmonary pressures (TPDP) might have received different treatments compared to those without TPDP measurements. The reviewer requested more information on the clinical interventions used to reduce TPDP and emphasized that presenting data on interventions is critical for clinical application.

Response: In response to the reviewer’s concern, we found that patients with decreased TPDP had a significantly lower mortality rate. It is important to emphasize that TPDP target is primarily a method for assessing respiratory mechanics rather than an intervention that directly impacts patient prognosis. We have revised the title and results sections to clarify the study's focus on target transpulmonary driving pressure (TPDP) values rather than the measurement of TPDP alone. This change aligns the title and text with the study's emphasis on target pressure values as the key influencing factors for ARDS outcomes. To investigate why patients with TPDP target had lower mortality, we examined the mortality differences and found that a TPDP cutoff value of 12.5 cmH2O was significant. Based on this finding, we grouped all patients who had TPDP target and observed that patients with TPDP < 12.5 cmH2O had a lower mortality rate. Notably, 78.3% of the patients in the TPDP-monitored group had TPDP values < =12.5 cmH2O, which accounted for a substantial proportion of the cohort. This suggests that the key factor contributing to the lower mortality in the TPDP-monitored group is effectively controlling TPDP below 12.5 cmH2O. We further investigated the specific parameters that could help achieve and maintain TPDP <= 12.5 cmH2O and found that effective control of peak airway pressure, plateau pressure, and driving pressure were critical in maintaining optimal TPDP values. Our mediation analysis also confirmed that peak airway pressure is an important mediator between TPDP > 12.5 cmH2O and increased mortality. To further substantiate these findings, we performed a separate analysis using propensity score matching (PSM) on patients with TPDP <= 12.5 cmH2O. After matching these patients to No-TPDP group with similar baseline characteristics, we observed that effective control of TPDP resulted in a significant reduction in mortality. In summary, while TPDP target serves as a method for assessing respiratory mechanics, the key to improving mortality outcomes lies in effectively controlling TPDP < 12.5 cmH2O. We hope these revisions enhance the clarity of the clinical implications of our findings. Thank you again for your thoughtful feedback. Please let us know if further clarifications are needed.

Table 1. The comparison between No-TPDP group and TPDP ≤ 12.5cmH2O group in matched cohort.

No-TPDP TPDP<=12.5cmH2O p

n 226 226

Age (median [IQR]) 58.35 [48.02, 68.80] 58.25 [43.12, 66.95] 0.497

BMI (median [IQR]) 31.40 [28.10, 36.80] 31.05 [26.90, 36.27] 0.372

APSIII (median [IQR]) 88.00 [63.00, 116.75] 88.00 [65.00, 106.00] 0.496

SOFA_score (median [IQR]) 8.95 [7.00, 11.78] 9.25 [7.03, 11.60] 0.833

ABPd (median [IQR]) 57.80 [54.45, 62.10] 58.15 [54.52, 62.18] 0.830

ABPs (median [IQR]) 108.45 [102.60, 114.72] 107.55 [101.95, 112.42] 0.329

Heart_Rate (median [IQR]) 92.40 [79.90, 105.00] 93.05 [80.32, 105.30] 0.810

WBC (median [IQR]) 12.95 [9.20, 16.35] 12.00 [7.93, 16.10] 0.213

Glucose (median [IQR]) 138.65 [111.70, 158.00] 137.15 [115.20, 163.65] 0.561

Creatinine (median [IQR]) 1.45 [0.90, 2.30] 1.30 [0.90, 2.18] 0.821

BUN (median [IQR]) 26.00 [17.85, 39.00] 23.25 [16.08, 35.25] 0.065

Lac (median [IQR]) 2.15 [1.60, 3.30] 2.25 [1.40, 3.38] 0.941

PlateletCount (median [IQR]) 178.20 [107.30, 246.25] 166.25 [110.32, 242.25] 0.672

PT (median [IQR]) 14.90 [13.50, 17.48] 14.75 [13.33, 17.62] 0.645

Arterial_PH (median [IQR]) 7.30 [7.30, 7.40] 7.30 [7.30, 7.40] 0.189

Arterial_O2_pressure (median [IQR]) 94.30 [79.80, 108.30] 93.30 [82.22, 108.52] 0.895

Arterial_CO2_Pressure (median [IQR]) 42.20 [37.00, 48.25] 42.75 [38.00, 48.10] 0.705

HCO3 (median [IQR]) 21.00 [18.00, 24.15] 20.80 [17.80, 23.95] 0.643

PFratio (median [IQR]) 91.85 [70.65, 133.72] 95.10 [70.17, 142.00] 0.694

Tidal_Volume (median [IQR]) 460.75 [409.52, 520.50] 427.20 [367.95, 474.82] <0.001

Spontaneous_Respiratory_Rate (median [IQR]) 0.30 [0.00, 3.08] 0.00 [0.00, 1.17] <0.001

Total_Respiratory_Rate (median [IQR]) 24.35 [20.30, 27.78] 26.05 [22.60, 28.87] 0.001

Set_Respiratory_Rate (median [IQR]) 22.70 [18.70, 26.00] 25.30 [21.60, 28.00] <0.001

PEEP (median [IQR]) 9.50 [7.20, 10.60] 10.60 [9.53, 12.00] <0.001

Plateau_Pressure (median [IQR]) 23.10 [19.80, 26.70] 26.35 [23.30, 29.08] <0.001

Peak_Pressure (median [IQR]) 26.35 [22.72, 30.87] 30.30 [26.22, 33.40] <0.001

Lung_compliance (median [IQR]) 33.75 [25.58, 43.08] 28.35 [22.63, 36.10] <0.001

Driving_pressure (median [IQR]) 13.75 [11.30, 16.17] 14.95 [12.10, 18.30] 0.002

Mechanical_power (median [IQR]) 20.35 [15.30, 26.58] 23.10 [18.30, 28.25] 0.002

Mechanical_ventilation_hour (median [IQR]) 83.05 [47.85, 141.62] 119.30 [71.72, 184.10] <0.001

ICUstayday (median [IQR]) 7.65 [3.80, 13.88] 10.70 [6.10, 19.62] <0.001

Hospitalstayday (median [IQR]) 13.15 [4.90, 21.20] 17.50 [8.83, 27.50] <0.001

28d_survival_day (median [IQR]) 28.00 [6.88, 28.00] 28.00 [16.55, 28.00] 0.029

28d_mortality(%) 87 (38.50) 66 (29.20) 0.037

ICUmortality (%) 84 (37.17) 64 (28.32) 0.045

2.Reviewer’ s comment: Page 2, Line 77: Use of the word "traditional"

Response: Thank you for the reviewer’s insightful comment. You are absolutely correct, and we acknowledge that we made a fault in referring to the "ARDSnet low tidal volume and plateau pressure control ventilation strategy from 2004" as the "traditional" lung-protective ventilation strategy. As you pointed out, this strategy differs significantly from the "traditional mechanical ventilation strategy," particularly with regard to the control of tidal volume and plateau pressure. We have revised the manuscript accordingly. We no longer refer to the 2004 ARDSnet strategy as "traditional." Instead, we now refer to it as the "lung-protective ventilation strategy (LPVS)." This change has been made in the revised manuscript, and we have updated the terminology to reflect this more accurate description. The revision can be found on page 2, line 68-71. Thank you again for pointing this out, and we hope this modification improves the clarity and accuracy of the manuscript.

3.Reviewer’ s comment: Page 3, Line 83: The following sentence is not clear. Please clarify: “However, the heterogeneity of disease manifestation among ARDS patients is ignored when only using lower tidal volume ventilation calculated by ideal body weight or limited platform pressure.” Do the authors mean that the current lung protective strategies do not take disease heterogeneity into account? And, what do the authors mean by disease heterogeneity? Severity, etiology or other?

Response: We appreciate the reviewer’s observation and have reworded this sentence for clarity. Our intention was to convey that with the growing understanding of ARDS pathophysiology and advancements in target techniques, using only tidal volume and plateau pressure control as the sole methods for lung-protective ventilation may not be beneficial for all patients. Specifically, such strategies may not adequately address the heterogeneity of ARDS patients. By heterogeneity, we mean the differences in etiology, pathology, biological characteristics, and treatment responses among ARDS patients. These differences highlight that a one-size-fits-all approach may not always lead to optimal outcomes. We have revised this part of the manuscript to more clearly express these points. The updated statement can be found on page 2, line 71-74. The revised sentence now reads: "However, current lung protective strategies, such as using lower tidal volume ventilation calculated by ideal body weight, fail to address the heterogeneity in disease manifestation among ARDS patients, which includes differences in disease severity and individual lung compliance[1-2]."Thank you again for your valuable feedback, and we hope this revision provides the necessary clarification.

[1] Deans KJ, Minneci PC, Cui X, et al. Mechanical ventilation in ARDS: One size does not fit all[J]. Crit Care Med, 2005, 33(5):1141-1143. 

[2] Goligher EC, Costa ELV, Yarnell CJ, et al. Effect of Lowering Vt on Mortality in Acute Respiratory Distress Syndrome Varies with Respiratory System Elastance[J]. Am J Respir Crit Care Med, 2021, 203(11):1378-1385.

4.Reviewer’s comment: Page 3, Line 87--A recent meta-analysis conducted through a randomized control trial by the authors found that mechanical ventilation 88 guided by TPDP was associated with decreased mortality among patients with ARDS.

Response: We apologize for our initial phrasing may have caused misunderstanding. The meta-analysis we referenced was published in 2023 and assessed the role and effects of mechanical ventilation guided by transpulmonary pressure (TPDP) in patients with ARDS. The meta-analysis included 13 randomized controlled trials (RCTs) that specifically investigated the use of TPP in ARDS patients. The revised manuscript now clearly reflects this, and the updated sentence can be found on page 3, line 77-80. The sentence has been revised to accurately reflect the methodology. It now reads: “A recent meta-analysis of randomized controlled trials conducted by the authors found that mechanical ventilation guided by transpulmonary pressure was associated with decreased mortality among patients with ARDS.” We sincerely apologize for the confusion caused by the previous wording and hope this revision resolves the issue.

5.Reviewer’s comment: Page 3, Line 90--Please reword the following to clarify the concept: Pirrone’ study found that end-expiratory transpulmonary pressure above 0 cm 90 H2O promoted collapsed alveolar reopen

Response: Thank you for the reviewer’s valuable comment. We apologize for the misunderstanding caused by our initial expression. We understand that our original phrasing was unclear and may have led to confusion. What we intended to convey is that Pirrone’s study found that during mechanical ventilation, target the end-expiratory transpulmonary pressure and adjusting parameters like PEEP to maintain the end-expiratory transpulmonary pressure above 0 cm H2O can help reopen collapsed alveoli, thereby improving lung compliance and oxygenation index. We sincerely apologize for the confusion caused by our previous wording, and we have revised the manuscript to clarify this point. The updated sentence can be found on page 3, line 80-82. The revised sentence now reads: ”Pirrone's study found that an end-expiratory transpulmonary pressure above 0 cm H2O facilitated the reopening of collapsed alveoli.”

6.Reviewer’s comment: Page 3, Line 95--Please provide a reference for this statement: However, excessive transpulmonary pressure may lead to alveolar overdistension, raising the occurrence of VILI.

Response: Thank you for the reviewer’s suggestion. We have added the following key references to this statement, covering the underlying mechanisms, animal studies, and clinical research, to strengthen the scientific basis. I have added the appropriate reference to support this statement: "Excessive transpulmonary pressure may lead to alveolar overdistension, thereby increasing the risk of ventilator-induced lung injury (VILI) [(1) Ball L, Talmor D, Pelosi P. Transpulmonary pressure target in critically ill patients: pros and cons. Crit Care. 2024 May 25;28(1):177. doi: 10.1186/s13054-024-04950-y; (2) Protti A, Andreis DT, Monti M, Santini A, Sparacino CC, Langer T, Votta E, Gatti S, Lombardi L, Leopardi O, Masson S, Cressoni M, Gattinoni L. Lung stress and strain during mechanical ventilation: any difference between statics and dynamics? Crit Care Med. 2013 Apr;41(4):1046-55. doi: 10.1097/CCM.0b013e31827417a6; (3) Sarge T, Loring SH, Yitsak-Sade M, Malhotra A, Novack V, Talmor D. Raising positive end-expiratory pressures in ARDS to achieve a positive transpulmonary pressure does not cause hemodynamic compromise. Intensive Care Med. 2014 Jan;40(1):126-8. doi: 10.1007/s00134-013-3127-1. Epub 2013 Oct 15. PMID: 24126673. ]."

7.Reviewer’ s comment: Can the authors please include the tests used to determine distribution of the data (e.g., normal vs skewed)?

Response: Thank you for the reviewer’s valuable feedback and for highlighting the need to include tests for data distribution. In response to the reviewer’s suggestion, we have now added a description of the tests used to assess the distribution of the data. Specifically, we used the Shapiro-Wilk test to test for normality. If the data followed a normal distribution, we used parametric tests. For non-normally distributed data, we used non-parametric tests. The updated sentence can be found on page 5, line 157-162.

Revised Text in the Statistical Analysis Section:

Descriptive Statistics

Normality was assessed using the Shapiro-Wilk test. Continuous variables following a normal distribution are expressed as mean ± standard deviation (MD ± SD), with comparisons between two groups conducted using the t-test. Non-normally distributed continuous variables are presented as median (interquartile range) [Median (IQR)], with comparisons performed using the Wilcoxon test. Categorical variables are expressed as percentages.

Table 2 Normality Test of variables in TPDP Group and No-TPDP Group.

Variable Test_Scope W_statistic P_value N

Age All 0.984 0.000 4721

Age No-TPDP group 0.983 0.000 4426

Age TPDP group 0.984 0.002 295

BMI All 0.899 0.000 4721

BMI No-TPDP group 0.902 0.000 4426

BMI TPDP group 0.907 0.000 295

SOFA_score All 0.968 0.000 4721

SOFA_score No-TPDP group 0.965 0.000 4426

SOFA_score TPDP group 0.988 0.012 295

APSIII All 0.980 0.000 4721

APSIII No-TPDP group 0.980 0.000 4426

APSIII TPDP group 0.990 0.047 295

ABPd All 0.983 0.000 4721

ABPd No-TPDP group 0.982 0.000 4426

ABPd TPDP group 0.987 0.008 295

ABPs All 0.962 0.000 4721

ABPs No-TPDP group 0.962 0.000 4426

ABPs TPDP group 0.967 0.000 295

Heart_Rate All 0.993 0.000 4721

Heart_Rate No-TPDP group 0.993 0.000 4426

Heart_Rate TPDP group 0.992 0.088 295

WBC All 0.990 0.000 4721

WBC No-TPDP group 0.990 0.000 4426

WBC TPDP group 0.988 0.019 295

Creatinine All 0.633 0.000 4721

Creatinine No-TPDP group 0.639 0.000 4426

Creatinine TPDP group 0.564 0.000 295

Glucose All 0.985 0.000 4721

Glucose No-TPDP group 0.985 0.000 4426

Glucose TPDP group 0.982 0.001 295

BUN All 0.900 0.000 4721

BUN No-TPDP group 0.899 0.000 4426

BUN TPDP group 0.916 0.000 295

PT All 0.920 0.000 4721

PT No-TPDP group 0.918 0.000 4426

PT TPDP group 0.949 0.000 295

PlateletCount All 0.967 0.000 4721

PlateletCount No-TPDP group 0.967 0.000 4426

PlateletCount TPDP group 0.967 0.000 295

Lac All 0.939 0.000 4721

Lac No-TPDP group 0.940 0.000 4426

Lac TPDP group 0.936 0.000 295

Arterial_PH All 0.854 0.000 4721

Arterial_PH No-TPDP group 0.850 0.000 4426

Arterial_PH TPDP group 0.836 0.000 295

Arterial_O2_pressure All 0.970 0.000 4721

Arterial_O2_pressure No-TPDP group 0.971 0.000 4426

Arterial_O2_pressure TPDP group 0.951 0.000 295

Arterial_CO2_Pressure All 0.993 0.000 4721

Arterial_CO2_Pressure No-TPDP group 0.993 0.000 4426

Arterial_CO2_Pressure TPDP

---

## [Decision Letter · Decision Letter 2]

Dear Dr. Fang,

Thank you for submitting your manuscript to PLOS ONE. After careful consideration, we feel that it has merit but does not fully meet PLOS ONE’s publication criteria as it currently stands. Therefore, we invite you to submit a revised version of the manuscript that addresses the points raised during the review process.

**ACADEMIC EDITOR:**

The manuscript presents important and timely findings that could contribute meaningfully to the field of critical care, particularly in advancing our understanding of transpulmonary driving pressure and its association with outcomes in ARDS. The authors have responded thoroughly to reviewer comments, and the manuscript is now technically sound, well-organized, and clearly written.

Both reviewers commend the improvements made and support the publication of this work **pending minor revision** .

**Required for Acceptance:**

This is the only substantive revision required for acceptance.

**Recommended (but not mandatory):**

No further changes are necessary beyond the clarification mentioned above. However, the authors may consider enhancing the clarity of their conclusions by briefly discussing potential future research directions, as implied by reviewer comments.

There are no conflicts between the reviewer recommendations. The authors are advised to revise the manuscript accordingly and resubmit for final evaluation.

I look forward to the revised version.

We look forward to receiving your revised manuscript.

Kind regards,

Gurmeet Singh, M.D., Ph.D.,

Academic Editor

PLOS ONE

Journal Requirements:

Reviewers' comments:

Reviewer's Responses to Questions

**Comments to the Author**

Reviewer #1: All comments have been addressed

Reviewer #3: All comments have been addressed

2. Is the manuscript technically sound, and do the data support the conclusions?

Reviewer #1: Yes

Reviewer #3: Yes

3. Has the statistical analysis been performed appropriately and rigorously?

Reviewer #1: Yes

Reviewer #3: Yes

4. Have the authors made all data underlying the findings in their manuscript fully available?

Reviewer #1: Yes

Reviewer #3: Yes

5. Is the manuscript presented in an intelligible fashion and written in standard English?

Reviewer #1: Yes

Reviewer #3: Yes

Reviewer #1: The authors have done an excellent job of addressing my questions and revising the manuscript. I would like to emphasize that evidence supporting the hypothesis about the effects of transpulmonary driving pressure in this manuscript are indirect. Proof of concept will require animal model experiments. The authors are obligated to mention this fact to the readers.

Reviewer #3: The authors have made significant improvements in addressing the concerns raised in the previous review. The manuscript is now technically sound, clearly written, and well-organized.

That said, I would like to emphasize that the conclusions drawn regarding transpulmonary driving pressure as a contributor to mortality are based on indirect evidence from observational data. While the findings support the hypothesis that elevated transpulmonary driving pressure is associated with higher mortality in ARDS, this does not constitute proof of causation. Proof of concept would require further validation, ideally through animal model experiments or prospective interventional trials.

I recommend the authors explicitly state this limitation in the Discussion or Conclusion section to ensure readers are aware of the inferential nature of these results. Therefore, I support publication pending minor revision to clarify this key point.

**Do you want your identity to be public for this peer review?** For information about this choice, including consent withdrawal, please see our Privacy Policy

Reviewer #1: **Yes: ** Mercedes Susan Mandell

Reviewer #3: **Yes: ** Aziza Harris

---

## [Author Response · Author response to Decision Letter 3]

10 May 2025

Dear Editors and Reviewers:

Thank you for your letter and for the Reviewers’ comments concerning our manuscript entitled “The Effect of Target Transpulmonary Driving Pressure Values on Mortality in ARDS Patients: A Retrospective Study Based on the MIMIC-IV Database”. (Submission ID PONE-D-24-38166R1). Those comments are all valuable and very helpful for revising and improving our paper, as well as the important guiding significance to our researches. We have studied comments carefully and have made correction which we hope meet with approval. Revised portion are marked in red in the paper. The main corrections in the paper and the responds to the reviewer's comments are as following:

Responses to the Reviewer 1’s comments:

1.The authors have done an excellent job of addressing my questions and revising the manuscript. I would like to emphasize that evidence supporting the hypothesis about the effects of transpulmonary driving pressure in this manuscript are indirect. Proof of concept will require animal model experiments. The authors are obligated to mention this fact to the readers.

Response: We sincerely appreciate your thoughtful and supportive comments on our revised manuscript. We are grateful for your recognition of our efforts in addressing your previous concerns. We acknowledge and fully agree with your important point that the current evidence supporting the role of transpulmonary driving pressure (TPDP) in ARDS is indirect and observational. As you rightly noted, proof-of-concept studies using animal models or prospective interventional trials are needed to establish a causal relationship. Accordingly, we have revised the Discussion section of the manuscript to explicitly state this limitation and emphasize the need for future mechanistic studies. Specifically, we have added the following sentence:

“It should be noted that our findings are derived from retrospective observational data, and the evidence supporting the role of transpulmonary driving pressure in influencing mortality is indirect. Causality cannot be inferred from these results. Therefore, proof of concept studies using animal models and prospective interventional trials are needed to validate the physiological mechanisms and causal links suggested by our findings.” The updated sentence can be found on page 22 line 419-425.

We hope this revision adequately addresses your concern and improves the clarity and scientific rigor of our manuscript.

Responses to the Reviewer 2’s comments:

1.The authors have made significant improvements in addressing the concerns raised in the previous review. The manuscript is now technically sound, clearly written, and well-organized. That said, I would like to emphasize that the conclusions drawn regarding transpulmonary driving pressure as a contributor to mortality are based on indirect evidence from observational data. While the findings support the hypothesis that elevated transpulmonary driving pressure is associated with higher mortality in ARDS, this does not constitute proof of causation. Proof of concept would require further validation, ideally through animal model experiments or prospective interventional trials. I recommend the authors explicitly state this limitation in the Discussion or Conclusion section to ensure readers are aware of the inferential nature of these results. Therefore, I support publication pending minor revision to clarify this key point.

Response: Thank you very much for your thoughtful and encouraging comments. We sincerely appreciate your recognition of the improvements made in the manuscript and your support for its publication. We fully agree with your observation that the association between elevated transpulmonary driving pressure (TPDP) and mortality in ARDS, as demonstrated in our study, is based on retrospective observational data and thus does not establish a causal relationship. As you rightly pointed out, proof of concept would require validation through animal models or prospective interventional trials. In response to your suggestion, we have revised the Discussion section of the manuscript to explicitly acknowledge this limitation. The following sentence has been added:

“It should be noted that our findings are derived from retrospective observational data, and the evidence supporting the role of transpulmonary driving pressure in influencing mortality is indirect. Causality cannot be inferred from these results. Therefore, proof of concept studies using animal models and prospective interventional trials are needed to validate the physiological mechanisms and causal links suggested by our findings.” The updated sentence can be found on page 22, line 419-425.

We hope our responses and revisions address all concerns raised by the reviewers. Please feel free to contact us if there are additional questions or suggestions. Thank you for your time and consideration.

Sincerely,

Mingxing Fang

m18533112886@hebmu.edu.cn

---

## [Decision Letter · Decision Letter 3]

The Effect of Target Transpulmonary Driving Pressure Values on Mortality in ARDS Patients: A Retrospective Study Based on the MIMIC-IV Database

PONE-D-24-38166R3

Dear Dr. Fang,

We’re pleased to inform you that your manuscript has been judged scientifically suitable for publication and will be formally accepted for publication once it meets all outstanding technical requirements.

Kind regards,

Gurmeet Singh, M.D., Ph.D.,

Academic Editor

PLOS ONE

Additional Editor Comments (optional):

Thank you to the authors for their careful and comprehensive revision of the manuscript entitled "The Effect of Target Transpulmonary Driving Pressure Values on Mortality in ARDS Patients: A Retrospective Study Based on the MIMIC-IV Database."

After re-evaluating the revised version and considering the reviewers' comments and the authors’ responses, I am pleased to recommend this manuscript for acceptance. The reasons are as follows:

Timely and Clinically Relevant Topic

The role of transpulmonary driving pressure in the management of ARDS is a highly relevant area of research, especially in the context of optimizing lung-protective ventilation strategies and minimizing ventilator-induced lung injury.

Robust Methodology and Analyses

The study utilizes a large, high-quality dataset (MIMIC-IV) and applies rigorous statistical techniques, including propensity score matching, causal mediation analysis, and phenotype-specific subgroup analysis, which enhance the internal validity of the findings.

Meaningful Clinical Insight

The identification of a TPDP threshold (>12.5 cmH₂O) associated with increased mortality provides valuable clinical insight that may inform individualized ventilatory management in ARDS patients.

Thoughtful and Adequate Revision

The authors have thoroughly addressed the reviewers’ concerns, particularly regarding the observational nature of the data and the need to clearly acknowledge the limitation of causal inference. The revised discussion now appropriately emphasizes this point.

Clear and Well-Written Manuscript

The manuscript is well-organized, scientifically sound, and clearly written, making it accessible to a broad readership, including clinicians and researchers in critical care and respiratory medicine.

Given the scientific merit, analytical rigor, and clarity of presentation, I believe this manuscript makes a valuable contribution to the literature and is suitable for publication in PLOS ONE.

Reviewers' comments:

Reviewer's Responses to Questions

**Comments to the Author**

Reviewer #3: (No Response)

2. Is the manuscript technically sound, and do the data support the conclusions?

Reviewer #3: Yes

3. Has the statistical analysis been performed appropriately and rigorously?

Reviewer #3: Yes

4. Have the authors made all data underlying the findings in their manuscript fully available?

Reviewer #3: Yes

5. Is the manuscript presented in an intelligible fashion and written in standard English?

Reviewer #3: Yes

Reviewer #3: I would like to thank the authors for their thoughtful and comprehensive revision of the manuscript. They have adequately addressed the concerns I raised in the previous review, especially regarding the observational nature of the findings and the need to clearly state the limitation of inferring causality.

The clarification added in the Discussion section (page 22, lines 419–425) effectively communicates this important point to the reader. The manuscript is now clear, well-organized, and technically sound. I have no further comments or concerns.

**Do you want your identity to be public for this peer review?** For information about this choice, including consent withdrawal, please see our Privacy Policy

Reviewer #3: **Yes: ** Aziza Harris

---

## [Editor Report · Acceptance letter]

PONE-D-24-38166R3

PLOS ONE

Dear Dr. Fang,

I'm pleased to inform you that your manuscript has been deemed suitable for publication in PLOS ONE. Congratulations! Your manuscript is now being handed over to our production team.

Kind regards,

on behalf of

Dr. Gurmeet Singh

Academic Editor

PLOS ONE